# Compliance with Infection Prevention and Control Practice among Prospective Graduates of Nursing School in South Korea

**DOI:** 10.3390/ijerph18052373

**Published:** 2021-03-01

**Authors:** Hyunjung Kim, Hyunju Park

**Affiliations:** 1College of Nursing & Research Institute of Nursing Science, Hallym University, Gangwon-do 24252, Korea; hjkim97@hallym.ac.kr; 2College of Nursing, Kangwon National University, Gangwon-do 24341, Korea

**Keywords:** compliance, infection control, nursing students, South Korea

## Abstract

The purpose of this study was to examine compliance with infection prevention and control practice and factors affecting compliance in nursing students who are about to graduate. A cross-sectional survey design was used. A total of 178 students from two nursing colleges in South Korea responded to self-reported questionnaires. Descriptive statistics, independent t-test, Pearson correlation, and multiple regression analysis were conducted. Mean score for compliance was 4.09 ± 0.43 out of 5. The highest score was on compliance with prevention of cross-infection (4.42 ± 0.54) and the lowest was on use of protective devices (3.86 ± 0.78). Non–occupational exposure scores on compliance subcategories were significantly higher than those for occupational exposure. Students’ perception of safe environment for infection control and a positive attitude toward infection control predicted compliance significantly (β = 0.28, *p* < 0.001; β = 0.18, *p* = 0.014, respectively). The findings clarify that the level of infection control compliance among Korean nursing students is moderate. In order to increase the level of compliance, a climate that emphasizes a safe environment for healthcare-associated infections should be strengthened. In addition, nursing education should endeavor to develop a positive attitude toward infection prevention.

## 1. Introduction

Healthcare-associated infections (HAIs) are considered one of the most important healthcare problems because they significantly affect the quality of healthcare and prognosis of patients [1]. According to the Centers for Disease Control and Prevention (CDC) in the United States, HAIs are steadily decreasing recently, but one in 25 hospitalized patients each year still experiences hospital infections [2]. The Ministry of Health and Welfare in South Korea estimates that 5–10% of hospitalized patients also suffer from HAIs; therefore, there is a need for prevention efforts against HAIs [3].

HAIs are caused by close contact between patients and healthcare workers. Among these workers, nurses spend the most time with the patients [4]. Nurses are therefore at the most risk of exposure to HAIs and can also cause cross-infection in patients. Nursing students are no exception [5,6]. They have direct contact with patients and perform diverse procedures that may contact the body fluids of the patient, in order to develop their skills during clinical practice training [7]. Thus, it is critical that nursing students should not to be exposed to HAIs.

Occupational exposure leading to HAIs is defined as when a nurse or nursing student is injured by a sharp instrument such as a needle or when the skin or mucosa are contaminated by blood, saliva, or other suspected infectious body fluids [8]. Previous studies have reported that nursing students have a high risk of occupational exposure to infections: varying from 8.8% to 73.0% according to geographical area [7,9,10]. About 23–73% of Korean nursing students have been reported to have experienced occupational exposure during clinical practice training, showing that efforts to reduce occupational exposure are necessary [11,12].

The best strategy for preventing occupational exposure is acknowledged to be raising compliance with infection prevention and control (IPC) practices such as standard precautions [13]. However, compliance with IPC practices has been evaluated mainly only in nurses, not nursing students, and nurses’ low compliance has been a constant problem [14]. Some recent studies have reported a similar trend of low compliance among nursing students [5,6,15]. In order to increase compliance, it is necessary to identify factors affecting it.

In order to examine compliance with IPC and its predictors among nursing students, it is best to study students who are about to graduate, to determine their level of compliance with IPC when they have almost completed their undergraduate program, as this also shows if programs need to be improved to raise compliance in students. To the best of our knowledge, although a few studies have evaluated compliance with IPC during clinical practice among graduating nursing students who have completed clinical practice [12,15], there are no studies evaluating occupational exposure as a factor affecting compliance with IPC. In this context, the purpose of this study was to evaluate compliance with IPC among prospective graduating nursing students and examine the factors influencing it. It aimed to comprehensively evaluate the relationships between occupational exposure, perceived safe environment, attitude toward and knowledge about IPC, and compliance with IPC. This study will inform compliance improvement efforts for Korean nursing students and point to ways to improve undergraduate education for prevention of occupational exposure.

## 2. Materials and Methods

### 2.1. Design and Setting

This study adapted a descriptive, cross-sectional study conducted at two university nursing schools in one city in South Korea. Since the university semester in Korea ends in December, the current study was conducted in December 2019, with 4th year students who were about to graduate.

### 2.2. Participants

This study used convenience sampling for prospective graduates from two schools of nursing. The inclusion criteria were that participants must be senior students who had completed clinical practice and who agreed to participate in the study. Using the G-Power program, 147 subjects were required for a medium effect size of 0.15, α = 0.05, power of 0.90, and 10 variables in the multiple regression analysis. The questionnaire was distributed to 200 students to compensate for the anticipated drop-out rate. The response rate was 89%.

### 2.3. Ethical Considerations

This study was conducted after approval by the institutional review board of the university with which the principal researcher was affiliated (IRB No. HIRB-2019-090). The respondents participated in the study after receiving explanation of the study’s goals and format and filling out the informed consent form. Research tools were modified and used after approval from the original authors.

### 2.4. Survey Tools

This self-administrated survey consisted of five parts. Part I collected demographic data such as age and gender, experience of and need for education related to IPC, and experience of exposure to needle injury and blood or body fluid. Part II assessed knowledge of IPC. A knowledge tool was developed based on the standard precautions against hospital-acquired infection of the Korea Center for Disease Control and Prevention [2], adapting the knowledge questionnaire developed by Mitchell et al. [15]. The tool consisted of 25 questions in three areas: standard precautions, transmission-based precautions, and multi-drug-resistant organism (MDRO) control. A correct answer was awarded 1 point and a wrong answer 0; the higher the score, the higher the level of knowledge. Part III evaluated attitude toward IPC using a tool consisting of nine items developed by Cho [16]. Each item was answered either yes (1) or no (0); higher scores indicated a positive attitude toward IPC. Part IV assessed students’ perception of the existence of a safe environment for IPC using the tool developed by Cho [16] and modified by Park [17], with nine items, answered yes (1) or no (0); the higher the score, the better the perceived environmental support for IPC. Part V measured students’ compliance with IPC during their clinical practice, using a tool modified by the present researcher from the compliance tool based on universal precautions developed by Regina et al. [18]. It consisted of 17 items using a 5-point Likert scale; the higher the score, the higher the compliance.

Internal reliability using the Kuder–Richardson Formula 20 (KR20) for the knowledge test was 0.64 in this study, while for attitude toward IPC it was 0.56. The KR20 for safe environment was 0.70 in Park [17] and 0.62 in this study. The compliance tool achieved a Cronbach’s alpha of 0.72 in the original study and of 0.71 in this study.

For construct validity of the compliance tool, the assumption of exploratory factor analysis (EFA) was satisfied based on the Kaiser–Meyer–Olkin index of 0.786 and the Chi-squared of 964.677 (*p* < 0.001) for Bartlett’s test of sphericity. The EFA supported five factors, following the original version of the tool, with cumulative variance percentage of 61.9%.

### 2.5. Data Collection

The researchers distributed questionnaires to the participants and explained the study’s goals, methods, potential benefits and risks, autonomy in participation, withdrawal from the study, and confidentiality. Subjects were also given an explanation of how to fill out questionnaires. Those who agreed to participate signed the consent form and completed the questionnaire at a convenient time. One week after the distribution of the survey, the consent form and questionnaire were collected through the student representatives so that the researchers could not influence the survey response.

### 2.6. Data Analysis

Data were analyzed using SPSS 23.0 (IBM Corp, Armonk, NY, USA). The participants’ background characteristics, experience of occupational exposure and the score of compliance with IPC were evaluated by descriptive statistics. The difference in IPC compliance score was examined through an independent t-test based on the experience of occupational exposure. The Pearson correlation was calculated to test the relationships between knowledge, attitude, safe environment, and compliance. A multiple regression analysis that included five factors (experience with needle injury, blood/body fluid exposure, knowledge, attitude, and safe environment) was conducted to identify predictors affecting compliance with IPC practices during clinical practice training. Reliability was evaluated by KR20 and Cronbach’s alpha. Construct validity for the compliance tool was evaluated using the EFA based on principal component analysis with varimax rotation.

## 3. Results

### 3.1. Personal Characteristics

Most of the participants were female (82.02%), with an average age of 22.75 (±2.28 years). While most participants (96.62%) were trained in IPC, more than half (63.48%) reported the requirement for further training on what has to be done after infection exposure. MDRO precaution was the second most frequently-mentioned area that required more training (48.31%). The attitude score toward IPC was 7.02 ± 1.20 out of 8.0, and perception of safe environment for IPC was 6.82 ± 1.75 out of 9.0. The knowledge score for IPC was 14.82 ± 2.12 out of 25, with 59.30% correct answers (Table 1).

### 3.2. Experience of Occupational Exposure during Clinical Practice

About 15% of participants had been exposed to needle injury during their clinical practice, and 49.44% of them had blood or body fluid exposure on their skin or mucous membrane. They were most exposed to blood (31.46%), followed by sweat (26.97%) and urine (20.22%). After occupational exposure, only 13.98% had reported to their clinical professors or head nurses. The reasons they did not report were that they had confirmed that the patient was not infected (50.00%) or believed the patient was not infected (51.25%); the total exceeds 100% due to multiple responses) (Table 2).

### 3.3. Compliance with Infection Prevention and Control Practice

The total mean score for compliance with IPC practices during clinical practice training was 4.09 ± 0.43 out of 5. Among the subcategories, the highest score was for compliance with cross-infection prevention practice (4.42 ± 0.54); in all, 87.64–93.82% of participants always or mostly performed all these items, including handwashing and glove changing. Use of protective devices showed the lowest score (3.86 ± 0.78): less than half (43.26%) of participants always or most often used goggles when they might be exposed to bloody discharge or fluid, the lowest compliance among protective device items (2.77 ± 1.60). Putting a dressing on one’s wound or lesion before patient care also showed relatively low compliance (3.69 ± 1.33). In the subcategory of decontamination of spills and used instruments, decontamination of surfaces and devices after use showed low compliance (3.72 ± 1.06) (Table 3).

### 3.4. Relationship between IPC Compliance and Occupational Exposure, Knowledge, Attitude, and Safe Environment

Participants who were occupationally exposed showed lower IPC compliance scores in some of the subcategories (Table 4). That is, students with needle injury experience had significantly lower compliance with the prevention of cross-infection (*p* = 0.043); furthermore those with exposure to blood or body fluid showed significantly lower compliance with the use of protective devices (*p* = 0.004) and the practice of decontamination of blood spills and used instruments (*p* = 0.022). 

The more positive the attitude toward IPC and the higher the perception of a safe environment for IPC, the higher the compliance with IPC practices (*p* for all < 0.001) (Table 5). However, no significant relationship was found between knowledge and the total compliance score (*p* = 0.456).

### 3.5. Predictors of Compliance with IPC Practices

The model significantly predicted compliance with IPC (*F* = 6.766, *p* < 0.001). Nursing students’ perceptions of a safe environment made a strong contribution to IPC compliance (β = 0.28, *p* < 0.001). Their attitudes toward IPC (*β* = *0*.18, *p* = 0.014) also made a significant contribution (Table 6).

## 4. Discussion

This study demonstrates that compliance with IPC practices among Korean nursing students ahead of graduation is moderate. The study participants (79.74%) showed higher compliance with IPC than students in Australia (59.8%) [15] and Saudi Arabia (61.8%) [6] but lower than in Jordan (84.3%) [19]. Such differences have been argued to be due to different curriculum and clinical environments among countries [19].

It is worth mentioning that not only this study but also previous studies have consistently reported low compliance with use of protective devices but high compliance with cross-infection prevention measures such as handwashing. In particular, the use of eye shields or goggles was the lowest among compliance areas investigated here, consistent with previous studies [6,20]. Wearing protective devices such as goggles, mask, and gown against blood or body fluids seems to be affected by their availability in the clinical setting. This is supported by a study reporting that compliance was improved by providing sufficient protection materials [21]. In this light, it is necessary to have sufficient protective devices and provide them to students in training hospitals. This will help prevent not only occupational exposure of vulnerable nursing students but also cross-infections in patients.

Another point to focus on is that, in this study, compliance with covering wounds or lesions with a waterproof dressing was also low, even though about half of students had had blood, urine, or sweat exposure on their skin or mucous membranes. Low compliance in covering one’s wound or lesion is consistent with studies by Alshammari et al. [6] (55.4%) and Colet et al. [22] (61.4%). These findings need to be noted because nursing students are vulnerable to occupational exposure to infection through their non-intact skin, if the patient has any infection. Hence, increasing students’ awareness about the necessity of dressing their wounds is important, as is an environment in which covering devices are readily available.

Meanwhile, as in this study, in previous studies few students officially reported their exposure to their clinical practice professors or preceptors for follow-up after exposure [7]. More than three-quarters of students did not know what to do after they were exposed in the study by Souza-Borges et al. [7], a finding supported by the finding in this study that 63.48% of students needed more education on how to cope after exposure. Nursing students should be trained to know about major pathogens that can infect them through occupational exposure, how to prevent such infections, how to cope with exposure, and how to report, before they begin their clinical practice training [7].

As well as blood and fluid exposure, 14.61% of nursing students had experienced needlestick injury during their clinical practice in the current study. In other studies, conducted in Korea [23] and Brazil [7], the incidence of needlestick injury among nursing students was as high as 26.9% and 67.6%, respectively. Although the rate of prior experience of needlestick injury was lower here than in the previous studies, it is still a serious problem in our context. According to previous studies, lack of technical skills and recapping of used needles are major reasons for needlestick injury [7,9]; however, universal precautions announced for IPC in 1987 indicate that used needles should not be recapped [24]. Despite continuing education about the risk of needle recapping, nursing students worldwide frequently recap used needles [9]. In this study, one-quarter of students recapped needles after giving an injection. Graduates of nursing school are novices or advanced beginners who need to practice developing their nursing skills. Therefore, preceptors should assess, educate, and monitor not only graduates’ skills but also their safety awareness when implementing procedures [9].

There was a significant association between occupational exposure and subscale scores for compliance, in spite of the lack of a significant relationship between occupational exposure and total compliance score. That is, students who had not experienced occupational exposure showed significantly higher compliance with IPC practices, including prevention of cross-infection, using protective devices, and decontamination of spills and used instruments, than those who had experienced it. This finding supports the assertion that improving compliance with IPC practices can be a good strategy to prevent occupational exposure to infection in nursing students [13,25].

In this study, perceived safe environment for IPC was a significant predictor of nursing students’ compliance with IPC practices, congruent with a study by Cruz [5] and with other studies conducted with Korean nurses [16,17]. Safe environment can be defined as the shared perception of management for safety support and feedback regarding IPC in hospitals, including a supportive work environment as well as adequate infrastructure and resources [14]. The fact that perception of a safe environment in relation to IPC affects students’ compliance means that the hospital’s infection prevention climate needs to be improved to increase students’ compliance and protect them and other health workers. To do this, administrative support is required [26]. In addition, other staff, in particular preceptors, in training hospitals need to actively work to prevent HAIs and help nursing students to perceive a safe environment as important.

The attitude score of students in the study was 7.02 out of 8, which is quite positive and was the other significant predictor for IPC compliance. In a study of knowledge and attitude in Jordanian nursing students, it was reported that not knowledge but attitude had a significant effect on compliance [27], which is the same result as in the current study. These results suggest that improving attitudes toward IPC practice is important. This is supported by a study on IPC practices among nurses, which showed that IPC compliance was motivated more by nurses’ subjective attitudes or beliefs than by objective knowledge [28]. For IPC compliance, knowledge is considered to be a mediator of behavioral change, causing a change in attitudes rather than directly changing behavior [12]. Therefore, it is necessary to increase compliance with IPC by raising awareness of the need for IPC and promoting a positive attitude.

Overall, this study has demonstrated that the participants have insufficient knowledge of IPC and that their knowledge was not significantly related to their compliance. Their low knowledge level was consistent with the 59.8% correct answer rate in a study that evaluated nursing students’ knowledge of standard precautions and transmission-based precautions [15]. However, it was lower than the 76.6–83.0% correct rate in studies that only assessed standard precautions [12,19]. Looking at previous studies, a few have found that knowledge has a significant effect on compliance [20], but many more studies have reported that knowledge has no significant effect [18,19,27,29]. It can be inferred that knowledge does not directly change practice and therefore does not affect compliance of nursing students [6]. In order to enhance compliance with IPC, it is necessary to improve educational methods that can be applied in clinical practice rather than simply conveying knowledge about IPC. According to the curriculum policy of nursing education in Korea, moreover, IPC education is provided as a small part of the fundamentals of nursing courses in the first or second year [12]. It mainly covers how to prevent patients’ infection in hospitals, and there is no curriculum dealing with students’ occupational exposure. It is necessary to continuously promote compliance with IPC through repetitive reinforcement during clinical practice courses rather than one-time education about IPC in a fundamentals of nursing course. Standardized guidelines for students’ occupational exposure need to be established and inculcated.

This study has several limitations. First, it may be difficult to generalize the findings of a study conducted at only two nursing schools in only one Korean city. Second, this descriptive study is based on self-reported compliance with IPC practice, which may be different from the situation found under direct observation. Third, since this study uses a cross-sectional design, it is not clear what the causal relationships are between independent and dependent variables. That is to say, although the non-occupational exposure group had a higher compliance score than the exposure group, it remains possible that highly compliant students are less likely to be occupationally exposed than less compliant students. Future studies that involve a large number of universities, assess compliance by other methods such as direct observation, and identify cause-and-effect relationships are warranted. Nevertheless, this study is significant for three reasons. First, the data for this research were collected when students were almost finished with the nursing program. Therefore, the paper was able to identify areas that required improvements to increase students’ compliance. Second, this is one of the few studies to have investigated the relationship between occupational exposure and compliance in nursing students. Finally, this study makes a contribution by indicating the importance of a perceived safe environment and of a positive attitude to improve nursing students’ compliance with IPC and prevent occupational exposure.

## 5. Conclusions

This study has highlighted that many students have experienced occupational exposure during clinical practice training. Improving compliance with IPC practice is a promising way to prevent occupational exposure. A safe environment improves compliance with IPC practice among Korean nursing students. It is necessary for teaching hospitals to establish a safe environment for IPC, including supportive work environment, infrastructure, and resources. Nurses, including preceptors, are role models for students in their practice settings, so they should be encouraged to improve their compliance. Nursing schools should incorporate up-to-date IPC policies such as precautions against MDRO into the undergraduate nursing curriculum and should educate students to have a positive attitude toward compliance with IPC through continuing education on the importance of IPC. It is essential to educate nursing students in how to protect themselves from occupational exposure related to HAI and how to cope with such exposure if it occurs. The efforts of nursing schools, teaching hospitals, and nursing students will hopefully lead to behavioral changes in relation to IPC among students.

## Figures and Tables

**Table 1 ijerph-18-02373-t001:** Background characteristics of participants (*n* = 178).

Characteristics	Category	*n* (%) or Mean (SD)	
Age in years		22.75	(2.28)
Gender	Male	32	(17.98)
	Female	146	(82.02)
Educational experience related to infection control			
Standard precaution	Yes	172	(96.62)
	No	6	(3.37)
Transmission-based precaution	Yes	167	(93.30)
	No	12	(6.70)
Multi-drug resistant organisms precaution	Yes	145	(81.46)
	No	33	(18.54)
Education needed for infection control *			
Standard precaution		38	(21.35)
Transmission-based precaution		61	(34.27)
Multi-drug resistant organisms precaution		86	(48.31)
Environmental management precaution		47	(26.40)
Coping methods after infection exposure		113	(63.48)
Attitude toward infection control	0–8	7.02	(1.20)
Perceived safe environment for infection control	0–9	6.82	(1.75)
Knowledge about infection control	0–25	14.82	(2.12)
Standard precaution	0–10	6.79	(1.08)
Transmission-based precaution	0–9	5.35	(1.42)
Multi-drug resistant organism control	0–6	2.68	(1.05)

* multiple responses.

**Table 2 ijerph-18-02373-t002:** Experience of occupational exposure during clinical practice (*n* = 178).

Variable	Category	*n*	(%)
Experience of needle injury	Yes	26	(14.61)
	No	152	(85.39)
Experience of blood or body fluid exposure in skin or mucus membrane	Yes *	88	(49.44)
	Blood	56	(31.46)
	Urine	36	(20.22)
	Stool	7	(3.93)
	Saliva	34	(19.10)
	Sweat	48	(26.97)
	Others	14	(7.87)
	No	90	(50.56)
Report to the clinical practice professor or head nurse (*n* = 93)	Yes	13	(13.98)
	No	80	(86.02)
Reasons not reported (*n* = 80) *	I confirmed that the patient was not infected	40	(50.00)
	I thought the patient would not be infected	41	(51.25)
	I did not know the reporting system	14	(17.50)
	I was afraid of the results after the report	4	(5.00)
	Others	7	(8.75)

* multiple responses.

**Table 3 ijerph-18-02373-t003:** Compliance with infection prevention and control practice (*n* = 178).

Item	Always or Most (%)	Mean (SD)
**Prevention of cross-infection**		4.42 (0.54)
I wash my hands before I take care of patients	93.82	4.35 (0.72)
I wash my hands after I take care of patients	91.57	4.39 (0.71)
I wash my hands immediately after removing medical gloves	93.26	4.54 (0.66)
I change gloves between patients	91.57	4.49 (0.76)
I wash my hands before nursing practice although I do not touch patients (e.g., medication)	87.64	4.34 (0.81)
**Use of protective devices**		3.86 (0.78)
I wear sterile surgical gloves when touching blood, body fluid, mucous membrane, or non-intact skin.	78.09	4.15 (1.00)
I cover my wound or lesion with waterproof dressing before caring of patients.	66.29	3.69 (1.33)
I wear protective gown if soiling with blood or body fluid is likely	79.22	4.11 (1.01)
I wear face mask when I am at risk of blood or body fluid splashed to my mouth	84.83	4.27 (0.81)
I wear eye shield/goggles when I may be exposed to the splashing of bloody discharge/fluid	43.26	2.77 (1.60)
I wear non-surgical gloves when I am exposed to blood or body fluids	79.22	4.20 (0.90)
**Decontamination of spills and used instruments**		3.99 (0.82)
I decontaminate surfaces and devices (e.g., thermometer, blood pressure manometer) after use	60.11	3.72 (1.06)
I clean up devices contaminated with blood using disinfectant	84.26	4.27 (0.99)
**Disposal of sharps**		3.97 (0.92)
I put used needles or scalpels in sharps box	98.88	4.83 (0.41)
The sharp box is only disposed of when it is full *	37.64	3.23 (1.44)
I recap needles after giving an injection *	25.84	3.87 (1.52)
**Disposal of waste**		4.39 (0.99)
Heavily bloodstained materials are packed in a medical waste container irrespective of patient’s infectious status	87.08	4.39 (0.99)
**Total**	79.74	4.09 (0.43)

* reverse coded for mean.

**Table 4 ijerph-18-02373-t004:** Differences in infection prevention and control (IPC) compliance according to occupational exposure (*n* = 178).

Compliance	Experience of Occupational Exposure					
	Needle Injury			Blood/Body Fluid Exposure		
	Yes	No	*t*(*p*)	Yes	No	*t*(*p*)
Total	3.95 ± 0.46	4.11 ± 0.43	1.803 (0.073)	4.04 ± 0.41	4.14 ± 0.45	1.553 (0.112)
Cross–infection	4.22 ± 0.58	4.46 ± 0.53	2.042 (0.043)	4.41 ± 0.50	4.43 ± 0.58	0.188 (0.851)
Protective device	3.70 ± 0.75	3.89 ± 0.78	1.172 (0.243)	3.69 ± 0.83	4.03 ± 0.69	2.956 (0.004)
Decontamination	3.98 ± 0.67	4.00 ± 0.85	0.091 (0.928)	3.85 ± 0.81	4.13 ± 0.81	2.305 (0.022)
Disposal of sharps	3.92 ± 0.87	3.98 ± 0.93	0.303 (0.762)	4.12 ± 0.88	3.83 ± 0.94	−2.189 (0.030)
Disposal of waste	4.15 ± 1.22	4.43 ± 0.95	1.334 (0.184)	4.42 ± 0.97	4.37 ± 1.02	−0.360 (0.719)

**Table 5 ijerph-18-02373-t005:** Relationships between knowledge, attitude, safe environment, and IPC compliance (*n* = 178).

	Knowledge	Attitude	Safe Environment
Compliance	*r*(*p*)	*r*(*p*)	*r*(*p*)
Total	0.056 (0.456)	0.283 (<0.001)	0.354 (<0.001)
Cross–infection	0.145 (0.053)	0.168 (0.025)	0.204 (0.006)
Protective device	−0.082 (0.279)	0.240 (0.001)	0.439 (<0.001)
Decontamination	0.037 (0.627)	0.077 (0.306)	0.183 (0.014)
Disposal of sharps	0.105 (0.164)	0.091 (0.229)	−0.110 (0.143)
Disposal of waste	0.055 (0.463)	0.144 (0.055)	0.018 (0.810)

**Table 6 ijerph-18-02373-t006:** Factors that impact on compliance with infection control practice (*n* = 178).

	*B*	*β*	*t*	*p*
Experience in needle injury exposure (yes)	−1.556	−0.074	−1.050	0.295
Experience in blood exposure (yes)	−0.328	−0.022	−0.305	0.761
Knowledge	0.066	0.019	0.267	0.789
Attitude	1.133	0.184	2.477	0.014
Safe environment	1.187	0.278	3.645	<0.001
*R*^2^ = 0.164, *F* = 6.766, *p* < 0.001				

## Data Availability

The data presented in this study are available on request from the corresponding author. The data are not publicly available due to privacy.

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
