# Peer review of "Compliance with Infection Prevention and Control Practice among Prospective Graduates of Nursing School in South Korea"

_ijerph, 2021, doi:10.3390/ijerph18052373_

Round 1

Reviewer 1 Report

The authors touched on a very interesting topic for academic institutions regarding student safety; specifically, compliance has already been the subject of other studies.

In line 55 authors said that "few studies evaluate compliance....". Which studies ? what do they claim and how do the authors differ in their purposes ? This is an issue that needs clarification.

Authors declared that "This study will inform compliance improvement efforts for Korean nursing students and point to ways to improve undergraduate education for prevention of occupational exposure among them". I do not find that the article specifically addresses this aspect because it remains on a general level that is independent of the contents and practices of those universities.

Moreover, in Table 1 - educational experience related to infection control - no questions were reported on behaviors to be implemented after exposure. I think this is an important point to understand. Were students provided with the necessary knowledge? Were the procedures clear?

The methods section I believe needs improvement. How were the analyses in Table 4 conducted? I do not understand....The same when I read section 3.4. lines 156-160.

In discussion I suggest staying more anchored to your context and thus to the goal that was stated. What concrete actions can you put in place to address the problems you have found in your settings?

Author Response

The revised parts of the paper are also marked in red.

Thank you.

Reviewer 2 Report

I found interesting to read the article.  Well organized. However, there are few issues and corrections that can be addressed to improve the article.

Line: 107: Who assisted participants to understand and fill out the questionnaire to avoid missing data

Line 121-124: Revise the sentence

Line 124: 7.02±1.20 (mean±SD), same for the perception of safe environment for IPC, and knowledge score

Line 132: Delete ‘had’

Line 147: Does the compliance with infection prevention and control practice have any cut-off value? (such as to determine low, medium, good or excellence) How to determine 3.69 is low or medium (what’s about 2.77).

Line 157: p = .043

Line 175: ‘overall’ compliance

Line 231: 178 students cannot represent Korean students’ characteristics, better to make comparison based on study to study (participants) such as study conducted in X country

Line 231-232: Better not to use Korean nursing students, just use study participants

Line 237-238: Making such statement, much evidence is needed. So, better to say the importance of attitude and safe environment without underestimating the knowledge score

Table 1: Background characteristics of participants

Age in years (table 1) or table 3: (% or SD) decimal number should be consistent in all tables (either 2.28 or 2.3)

Author Response

(The authors gave the same response as above.)

Reviewer 3 Report

The presented work, although relates to public health seems to be a bit beyond the scope of IJERPH.  Authors put an effort into assessing what should be the basis of the correct education of nursing students. I agree that  it is necessary for teaching hospitals to establish a safe environment for IPC, including supportive work environment, infrastructure, and resources. But is it not obligatory in South Korea? How any kinds of classes, practical classes, can start without establishing and providing safe working conditions? Why students need to be educated to have a “positive attitude toward compliance with IPC”? Is it not a work of the instructors, school administration to improve compliance? What I saw in table 3 scared me. The value below 100% in the second column shows that there is insufficient education, or even lack of professional qualification, and importantly testifies to insufficient control of sanitary conditions.

Authors admit “this study has demonstrated that Korean nursing students have insufficient knowledge of IPC and that their knowledge was not significantly related to their compliance”.

The basic question that should be included in the paper is what universities/medical schools do to better educate their responsible students? Unfortunately, this is not presented in this work. There is a diagnosis or a presentation of  the very bad state data, but no explanation as to what led to it. The current policy of nursing education schools should be more discussed.

Author Response

(The authors gave the same response as above.)

Round 2

Reviewer 1 Report

The authors have taken the suggested elements into account and the text has been improved. 

Some minor consideration:

line 185...Australia..not Australian

I think the text given in lines 175-177 .....is more appropriate to put it in the methods.

Thank you for your effort.

Author Response

Dear reviewer, 

All changes were made in blue in the manuscript.

line 185...Australia..not Australian

  • Response 1: We changed “Australian’ to ‘Australia’.

I think the text given in lines 175-177 .....is more appropriate to put it in the methods.

  • Response 2: We put the sentences in lines 175-177 in the methods (at the section 2.6)